# Implementation, delivery, and utilization of iron fortified rice supplied through public distribution system across different states in India: An exploratory mixed-method study

E. R. Nandeep[1⊙], Hemant Mahajan[2⊙], Mahesh Kumar Mummadi[1], Challa Sairam[2], Venkatesh K.[3], Jayachandra Kadiyam[2], Indrapal Meshram[4], Sreenu Pagidoju[1], Venkata Raji Reddy[1], Hrusikesh Panda[5], Raghu Pullakandham[6], J. J. Babu Geddam[1], Subbarao M. Gavaravarapu[5], Hemalatha R.[7], Samarasimha Reddy N.[1]*

1 Clinical Epidemiology Division, ICMR-National Institute of Nutrition, Beside Tarnaka Metro Station, Hyderabad, India, 2 Maternal and Child Health Nutrition Division, ICMR-National Institute of Nutrition, Beside Tarnaka Metro Station, Hyderabad, India, 3 Extension and Training Division, ICMR-National Institute of Nutrition, Beside Tarnaka Metro Station, Hyderabad, India, 4 Public Health Nutrition Division, ICMR-National Institute of Nutrition, Beside Tarnaka Metro Station, Hyderabad, India, 5 Nutrition, Information, Communication and Health Education Division, ICMR-National Institute of Nutrition, Beside Tarnaka Metro Station, Hyderabad, India, 6 Drug safety and Toxicology Division, ICMR-National Institute of Nutrition, Beside Tarnaka Metro Station, Hyderabad, India, 7 Director, ICMR-National Institute of Nutrition, Beside Tarnaka Metro Station, Hyderabad, India

⊙ These authors contributed equally to this work.
* drsamarareddy@gmail.com

**Data Availability Statement:** The data collected for the study include sensitive information about

## Abstract

Food fortification with micronutrients is one of the cost-effective and sustainable methods to prevent micronutrient deficiencies at community level. The rice fortified with iron, folic acid, and vitamin B12 is being supplied through various social welfare schemes in India in a phased manner and planned to cover the entire country by March 2024. We have conducted a situational analysis to assess the rollout of fortified rice supplied through the Public Distribution System (PDS) and to evaluate the accessibility, availability, acceptability, and utilization of fortified rice by the beneficiaries of the PDS. This was a mixed-method, sequential exploratory study conducted in six districts from six different states of India that had begun distribution of fortified rice through PDS in pilot mode during 2020–2021. In each district, the district supply officer of the PDS, Food Corporation of India (FCI) or State Food Corporation (SFC) warehouse supervisor, and four Fair Price Shop (FPS) dealers were interviewed. Under each FPS, a minimum of seven beneficiary households were randomly selected and interviewed using a structured questionnaire. The in-depth interviews were conducted with different stakeholders using theme guides. All the interviewed stakeholders were aware about their roles and responsibilities and purpose to distribute fortified rice. There was a continuous supply of fortified rice (across all visited districts) to beneficiaries through a well-established system. Acceptability and compliance to intake of fortified rice was good with no reported gastrointestinal adverse outcomes following fortified rice intake. There was an efficient roll-out of fortified rice though PDS with a good compliance to intake of fortified rice. It

participants (district supply officers, fair price shop dealers and beneficiaries of public distribution system) and their opinion towards the government's policies and programs and various welfare schemes/services. Despite deidentification of the data, markers can be picked up from the transcripts through which the identity of the participants may be compromised. Therefore, sharing of data publicly will not be appropriate, as this information will put the participants at risk of being identified. Additionally, the institutional ethics committee of the Indian Council of Medical Research – National institute of Nutrition, Hyderabad, India doesn't allow publicly sharing of data for publication purpose. However, there is a provision for the bonafide researchers to access the de-identified data after seeking prior approvals from the Principal Investigators (Dr. Samarasimha Reddy N; Email: drsamarareddy@gmail.com / Dr. Hemant Mahajan; Email: Hemant. mahajan.84@gmail.com) and the Member secretary of the ethics committee Indian Institute of Public health, Hyderabad, India (Dr. Uday Kumar, Email ID: putchaaudaykumar@yahoo. com).

**Funding:** The authors received no specific funding for this work.

**Competing interests:** The authors have declared that no competing interest exist.

is feasible to design and conduct a study to assess the impact of fortified rice on anemia and iron storage at the community level.

## Introduction

Anemia, defined as the hemoglobin levels below an age and gender specific threshold [1], commonly presents with symptoms mostly related to the reduced supply of oxygen to the tissues such as fatigue, shortness of breath, lethargy, tiredness, poor mental and scholastic performance, and cold intolerance [2,3]. Anemia poses a major public health concern globally with a prevalence of 24.3% across all ages [4] and especially in developing countries including India [5]. The fifth National Family Health Survey (NFHS-5) survey conducted in India between 2019 and 2021 showed a very high prevalence of anemia across all the vulnerable age groups: children aged 6–59 months (67%); women of reproductive age (57%), and pregnant women (52%) [6]. The Comprehensive National Nutrition Survey (CNNS) conducted in 2016–2018 found the prevalence of anemia among children aged 1–4 years as 40.5%, while it is 23.5% among children aged 5–9 years and 28.4% among adolescents [7]. Although anemia is multifactorial, iron deficiency anemia is the most common form of nutritional anemia, which contributes majorly to the global anemia prevalence [8,9].

India has taken multiple initiatives to reduce the burden of anemia at community level [10]. The 'Anemia Mukt Bharat' (Anemia free India) program envisages a 6x6x6 strategy with six beneficiaries, six interventions, and six institutional mechanisms, to reduce the prevalence of anemia [11]. One of the six key interventions among them is the mandatory provision of iron and folic acid fortified foods distributed through the public health programs [11]. Fortification of rice with folic acid, in addition to preventing megaloblastic anemia, could have the additional benefit for protecting against the development of neural tube defects by delivering the vitamin to women before and during pregnancy [12–16].

As a part of the 'Anemia Mukt Bharat', the country has planned to supply fortified rice, fortified using rice kernels containing micronutrients such as iron (28–42.5 milligram (mg) per kilogram (kg) of rice), folic acid (75–125 microgram per kilogram of rice), and vitamin B12 (0.75–1.25 microgram per kilogram of rice) through various social welfare schemes throughout the country in a phased manner by March 2024 [17]. In the phase-I of the roll out, the fortified rice was introduced in the social welfare schemes such as Integrated Child Development Scheme (ICDS) and Pradhan Mantri Poshan Shakti Nirman (PM POSHAN, earlier known as the National Program of Mid-Day Meal in Schools) throughout India during 2021–22 [18]. Phase-II has covered aspirational and high burden districts for anemia (total 291 districts) under Public Distribution System (PDS) and other welfare schemes, in addition to Phase-I districts, by March 2023 [18]. All the remaining districts in India will be covered in Phase III by March 2024 [19]. Public Distribution system is a supplemental food security system for providing foodgrains at affordable prices in India. They provide rice, wheat, sugar, and kerosene. In some states, other items such as pulses, iodized salt, and edible oil are also provided. These commodities are provided through the Fair Price Shops (FPS), which are also known as Ration shops [20].

Food fortification is considered a tested, cost-effective, and harmless method to prevent nutrient deficiencies especially when staple foods are fortified [21–24]. A Cochrane review on rice fortification with iron alone or in combination with other micronutrients analyzed seven Randomized Control Trials (RCTs) with a total of 1634 participants and found that the risk of anemia with unfortified rice was 388 per 1000, while with fortified rice with iron alone or in

combination with other micronutrients was 279 per 1000, resulting in an absolute reduction of 109 per 1000 (risk ratio (RR) 0.72, 95% confidence interval (CI) 0.54 to 0.97; $I^2$ = 74%) [25]. However, the review classified the evidence as 'low-certainty'. The review also found that the fortification of rice with iron (alone or in combination with other micronutrients) may reduce the risk of iron deficiency although the effect on anemia remained equivocal. Studies on Indian children showed varying results on the prevalence of iron deficiency anemia and body iron stores with the consumption of fortified rice [26,27].

It is crucial to gain insights of the functioning and utilization of existing scheme/provisions and related issues with scaling up of fortified rice distribution to all the remaining districts of India. In this view, a mixed-method study was conducted in six pilot districts with fortified rice distribution through PDS with the objective to understand the rollout of fortified rice supplied through the PDS and to evaluate the accessibility, availability, acceptability, and utilization of fortified rice by the beneficiaries of the PDS. The information generated through this situational analysis will help us to design a study to evaluate the effectiveness of fortified rice on anemia and iron storage at community level.

## Materials and methods

### Study design and approvals

The study employed a mixed-method, sequential exploratory design for collecting quantitative and qualitative data from different stakeholders on supply/distribution and consumption sides. The quantitative and qualitative data were triangulated to get a comprehensive picture and to avoid flaws and possible research bias of relying on a single method of data collection. An ethical committee approval was taken from the Institutional Ethical Committee of Indian Council of Medical Research (ICMR)- National Institute of Nutrition (02/II/2023). The list of the districts with fortified rice distribution through PDS as a pilot project was obtained from the Ministry of Civil Supplies, Government of India (S1 Table). From this list of districts, six districts were randomly selected for the study. Necessary approvals were taken from Ministry of Civil Supplies, Government of India, and the respective State governments for conducting the study. Written informed consents were taken from all the stakeholders and beneficiaries before the interview. Prior Consent was taken from all the stakeholders for audio-recording the interviews.

### Study setting

We had obtained the list of the districts (S1 Table) from the Ministry of Civil Supplies, Government of India (GOI), where the fortified rice rollout through PDS as a pilot project had begun (2020–2021). From this list, we randomly selected six districts from six different states of India for our study: Gadchiroli in Maharashtra, Narmada in Gujarat, Vijayanagaram in Andhra Pradesh, Tiruchirappalli in Tamil Nadu, Chandauli in Uttar Pradesh, and East Singhbhum in Jharkhand. In each district, four fair price shops (FPS) from different sub-districts were randomly selected with an urban to rural ratio of 1:3.

### Study implementation

A mixed methods study involving different stakeholders and beneficiaries of the public distribution system (PDS) was conducted from May 1 to 5, 2023 in the selected six districts. Quantitative data were collected from beneficiaries of the program from the consumption side. In each district, the district supply officer of the public distribution system was interviewed to understand the rollout of the program, Godown (Warehouse) Managers of Food Corporation of India (FCI) and the State Food Corporation (SFC) were interviewed to understand the

**Table 1. List of stakeholders interviewed and places visited.**

| State | District | Total Stakeholder Interviews |
|---|---|---|
| Andhra Pradesh | Vijayanagaram | • 13 District Officials |
| Gujarat | Narmada | • 13 Warehouse managers/supervisors |
| Jharkhand | East Singhbhum | • 26 FPS Dealers |
| Maharashtra | Gadchiroli | • 181 beneficiaries |
| Tamil Nadu | Tiruchirappalli | |
| Uttar Pradesh | Chandauli | |

supply, storage, and transport of fortified rice. Four fair price shop (FPS) dealers in each district were interviewed to understand the distribution of fortified rice to the beneficiaries. All these interviews were conducted using content validated, pre-tested, closed-ended questionnaires. We selected these diverse group of stakeholders with the prior assumption that they share the common characteristics of serving community individuals and have the potential to provide relevant and diverse information for the study objectives. We also conducted the in-depth interviews with structured questionnaire for a minimum of seven beneficiary households per FPS. A total of 181 beneficiaries were interviewed from six districts to retrieve diverse and comprehensive information (Table 1). The beneficiaries provided quantitative responses to the structured interview questionnaire, while in-depth interviews were conducted with the other stakeholders.

In consultation with nutrition experts, we developed interview questions and probes (interview guides) with special focus to understand: (i) Stakeholders' perception about the roll out and distribution of fortified rice through various welfare schemes; (ii) The factors (context, experience, and skills of DSO, warehouse supervisors, FPS owners, operational management, infrastructure, geography, beneficiary attitude and perceptions) that facilitated or hindered implementation and acceptability of fortified rice through various schemes; (iii) The mechanisms of supply, storage, and transport of fortified rice; (iv) The level of access, availability, and utilization of fortified rice supplied through PDS by the beneficiaries of the program. (v) Reasons of non-compliance to fortified rice, and system level barriers to access PDS services.

Investigators with specialization in public health had conducted physical interviews using paper-based forms. The interviewers were not related to any of the interviewed participants. All interviewers had undergone a standardized training at the ICMR-National Institute of Nutrition-Hyderabad, India.

Qualitative data were collected by conducting in-depth interviews (IDIs) with different stakeholders using theme guides. Prior to conducting interview with any of the stakeholder, the interviewer completely explained the aims and objectives of the study and need for the interview recording, and clarified the participants' queries (if there were any) to seek an informed written consent. At the time of each interview, interviewer made sure that the participants were alone in the room. Each interview (for district officials, FPS dealer, FCI/SFC manager, as well as beneficiaries) lasted for 35 to 40 minutes. The IDIs were conducted by a team of trained interviewer and note-takers in each location in a standardized method (17). Each interview was audio-recorded with the prior permission of the interviewee and transcribed into word files. During the qualitative data collection, adequate care was taken not to include those who participated in the quantitative study.

## Data analyses

For the analysis of qualitative data, we transcribed the interviews and translated to English on the same day using notes taken by the interviewers to supplement the audio recordings. We

used a grounded theory approach to manually code the data. Open coding was done initially to look for distinct concepts and categories in the data to form the core themes of analysis. We then did axial coding by reading the transcripts again using the codes developed prior and classified statements based on the themes. Further, we arranged these codes as branching sub-themes. We marked the verbatims from the transcripts that aligned with the themes and the quotes that were completely divergent from others and were included in result section to improve the transparency of the findings. We collected the beneficiary data using a structured questionnaire and we entered and analyzed them using Microsoft Excel 2019 software.

## Results

A total of 217 interviews were conducted, including 52 unstructured and 181 structured in-depth interviews. Qualitative data analyses revealed different overarching themes from the in-depth interviews of district supply officers, warehouse supervisors, and FPS dealers with some overlap. The themes broadly described the various aspects related to the access, availability, storage, supply chain mechanism, and utilization of fortified rice supplied through PDS through the program. Some of the findings were as follows:

### 1. District supply officer

**Theme 1 –Acquisition of rice for a given district.**   Two prominent methods of rice acquisition for the selected districts were reported in the interviews of DSOs: (i) State government procuring paddy directly from farmers and fortifying rice with iron and other nutrients; (ii) State governments receiving fortified rice directly from the FCI warehouses;

"Paddy is directly procured from the farmers by the district administration at the government designated centers throughout the district at the minimum support price and supplied to authorized millers having facilities (blenders) for mixing raw rice with FRK." (DSO, Gadchiroli district, Maharashtra)

"The fortified rice comes from the FCI warehouses to the state warehouses which are present in each taluka of the district. The rice comes once every month to state warehouses from the FCI warehouses." (DSO, Narmada District, Gujarat)

**Theme 2 –Distribution of rice.**   Fortified rice distribution followed nearly the same pattern in almost all the states. The rice goes from the FCI warehouses to the state warehouses first, and then it is distributed to various FPSs. However, in Uttar Pradesh, it followed a different order as reported below:

"Distributors in the FCI are provided with distribution plan across the FPS of the district based on the block wise allotment. The truck goes from FCI Warehouse to the FPS tracked by GPS. DSO gives the schedule of distribution to the FPS. One nodal officer sits at FPS and monitors the distribution. Biometric system is being used for fool proof distribution at the FPS to the beneficiary." (DSO, Chandauli district, Uttar Pradesh) (Fig 1)

The farmer supplies raw rice to Paddy Procurement Centers. The food corporation of India (FCI) is involved in the performance appraisal of paddy procurement centers. PPCs stack the paddy. Packed paddy is sent in trucks to Rice mills for hulling of paddy. During the hulling, fortified rice kernels are mixed with the regular rice in the ratio of 1:100 (FRK: Regular rice grains). Mixed rice 50 kg jute bags are delivered from the rice mills to FCI Depots /Warehouses. FCI depot does the analysis of rice by lot quality and stores the rice. From here, rice is

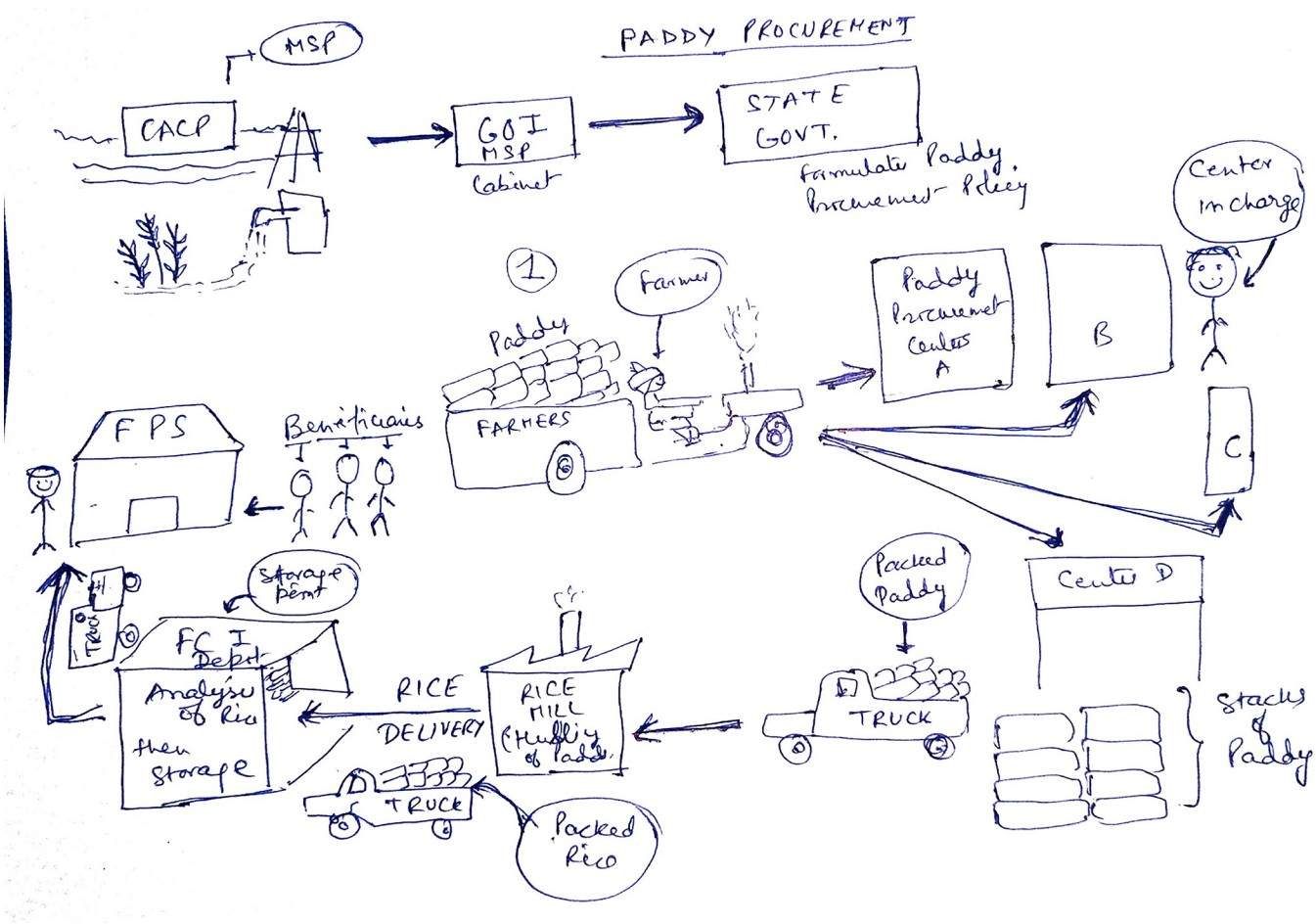

**Fig 1. Illustration of the FRK rice life cycle by District Food Marketing Officer (DFMO), Chandauli, Uttar Pradesh.**

directly transferred to fair price shops. Distributors in the FCI are provided with distribution plan across the FPS of the district based on the block wise allotment. The truck goes from FCI Warehouse to the FPS tracked by GPS. The UPePDS is a fool-proof system with secure Biometry-based distribution that maintains a real-time record of collection and demand of the FRK Rice. District supply officer gives the schedule of distribution to the FPS. One nodal officer sits at FPS and monitors the distribution. Biometric system is used for fool proof distribution at the FPS to the beneficiary.

CACP, commission for agricultural costs and prices; Dept, department; FPS, fair price shop; GOI: government of India; GOVT, government; MSP, minimum support price.

In certain states like Jharkhand, FRKs are sent to laboratories for testing the levels of fortificants before the rice is distributed:

"The miller sends FRK to the laboratory situated at Ranchi. Only after getting the approval certificate, fortified rice is distributed." (DSO, East Singhbhum District, Jharkhand)

**Theme 3 –Quality assessment.** The District Supply Officers in almost all states except Uttar Pradesh were not very aware of the intricate details related to the Quality Assessment of the fortified rice.

"Only manual quality check of rice is being done from our side." (DSO, Narmada District, Gujarat)

"GOI (Government of India) issues guidelines periodically on Quality Control (QC). Food safety licenses are monitored. Quality controller shares the list of empanelled Government and Private Labs and samples are sent for QC at 3 levels-

FRK Manufacturer: FRK premix and FRK

Rice Miller: FRK Rice for Distribution

FCI Warehouse: FRK Rice for Distribution."

(DSO, Chandauli District, Uttar Pradesh)

**Theme 4 –Feedback from the public related to distribution of fortified rice.** All DSOs gave us a similar response about the feedback from the beneficiaries. They reported that initially there was resistance from the public to using the fortified rice. But, various Information-Education-Communication (IEC) activities carried out by them helped remove the misconceptions and now the feedback is all positive. The IEC activities included posters and banners in FPS shops, demonstrations with live experiments to remove the myth of "plastic rice" by burning the rice, interactive talks carried out by experts (for example, experts from Food Research Laboratory and Nutrition International in Gujarat).

"We even carried rice cookers and demonstrated and consumed fortified rice along with beneficiaries. Activities like this led to a gradual acceptance of fortified rice among the public." (DSO, Vijayanagaram District, Andhra Pradesh)

"Earlier there was a misconception about fortified rice being slightly pink in color, floating while cooking, called it plastic rice, rice from China, etc. Beneficiaries used to fight with FPS owners as well previously with the same complaints. Now, it is all good feedback because of IEC activities at the village level." (DSO, Narmada District, Gujarat)

"80% of the public in the urban region and 95% of the public in the rural region are consuming fortified rice supplied through PDS. It took six months for the public to understand and use fortified rice." (DSO, East Singhbhum District, Jharkhand)

**Theme 5 –Alternative mechanisms to procure the fortified rice.** There are some additional themes that appeared in certain states, such as contingency mechanisms in Gujarat and the need for an impact assessment reported in Uttar Pradesh.

"If fortified rice is not available in our district, it will be taken from the neighboring secondary base depots at Barod and Chhani." (DSO, Narmada District, Gujarat)

"There is a need for impact study as we feel that we have been implementing a fairly well-designed program and we will be happy to know its impact." (DSO, Chandauli, Uttar Pradesh)

## 2. FCI Warehouse supervisor

**Theme 1 –Procurement of rice for a district.** The mechanism of procurement of rice, as narrated by the FCI Warehouse Supervisors, were aligning with what the DSOs mentioned about the acquisition of rice. The flow of procurement of rice differs from state to state,

although it can be broadly categorized into two: from different parts of the country to the non-paddy producing states, and from the state's paddy procurement centers in majorly paddy cultivating states.

> "The farmer supplies the raw rice to Paddy Procurement Centers (PPC). FCI is involved in the performance appraisal of paddy procurement centers. PPCs stack the paddy. Packed paddy is sent in trucks to Rice mills for hulling of paddy. During hulling, 100:1 (Rice: FRK) rice 50 kg jute bags are delivered from the rice mills to FCI Depots /Warehouses. FCI Depot does the analysis of rice by lot quality and stores the rice. From here rice is transferred to FPS directly." (a Warehouse Supervisor, Uttar Pradesh)

> "FCI Warehouses receive rice from different parts of the country in trains. The Warehouse located in Jamshedpur has a huge storage capacity and rice gets stored there based on the releasing order received from the state govt., Fortified rice will be transported in the lorries to the SFC Warehouses." (An FCI Warehouse Supervisor, Jharkhand)

**Theme 2 –Supply of rice from FCI Warehouses.** The FCI Warehouses supply the rice to the state warehouses/godowns from where the rice is supplied to the various FPS. In some of the states, there was also Global Positioning System (GPS) tracking of these trucks that carry the loads to prevent any malpractices.

> "This warehouse supplies to two districts in Gujarat: Narmada and Bharuch. The FCI sacks are not exactly 50kg. In the State Civil Supplies Warehouses, they make it exactly 50kg of fortified rice and then supply to respective FPS." (An FCI Warehouse Supervisor, Gujarat)

> "The quantity of rice supplied to each PDS shop is prefixed by the Commissioner of Civil Supplies based on the number and types of cards attached to each shop. Sixty percent of rice for the following month is supplied from the 20th to the 30th of the previous month (Advanced Movement). The remaining 40% is supplied from the 1st to 20th of the month (Regular movement)." (An FCI Warehouse Supervisor, Tamil Nadu)

**Theme 3 –Storage and quality check.** There were written protocols for storage of the fortified rice and all warehouses were found to follow them. They religiously followed the FIFO (First-In First-Out) policy as well—the stock that comes first, goes first.

> "A dedicated team of officers conducts quality assessment of stored rice on monthly basis. They check parameters such as legible stenciling/tagging, weighing of rice consignment, moisture content, percentage of broken grains cracked grains, immature grains, rice breakability, discolored/fermented grains, level of foreign material and infestation." (An FCI Warehouse supervisor, Maharashtra)

> "Periodically, prophylactic and curative treatments are being given at buffer warehouses and FCI warehouses. The disinfectant protocols are being followed at warehouses as prescribed by FCI, and no special measure is being taken for fortified rice. In buffer warehouses, we randomly collected 250g of rice, from which 50g of rice will be tested for FRKs. If this does not match 0.8 to 1.2%, the rice is rejected and sent back to the miller." (FCI Warehouse Supervisor, Andhra Pradesh)

**Theme 4 –Alternative mechanisms for fortified rice storage.** We received some suggestions and feedback from the FCI warehouse supervisors:

"A color-based detection kit to count the FRKs to check the quality is needed, as we miss some FRKs when done manually." (An FCI Warehouse Supervisor, Andhra Pradesh)

"I feel that the fortified rice should be distributed very fast as FRKs will start dissolving and become chalky. The rice type supplied in Jharkhand state is parboiled." (An FCI Warehouse Supervisor, Jharkhand)

### 3. Fair price shop dealers

**Theme 1 –Procurement and distribution of rice.** FPSs were the point of contact of rice with the beneficiaries. The FPSs were usually fixed shops where the beneficiaries come to get the fortified rice and other commodities. However, in Andhra Pradesh, they had Mobile Dispensing Units. Technology and GPS are also put to good use for the smooth functioning of the supply chain and distribution.

"The beneficiaries are receiving fortified rice through mobile dispensing units (MDUs) with Aadhar verification at their doorsteps. Hence, there are no issues related to distribution of fortified rice." (An FPS Dealer, Vijayanagaram District, Andhra Pradesh)

"Uttar Pradesh Electronic Public Distribution System (UPePDS) is a foolproof system with secure Biometry-based distribution that maintains a real-time record of collection and demand of the FRK Rice." (An FPS dealer, Chandauli district, Uttar Pradesh)

**Theme 2 –Perceived difference in quality of fortified rice.** Some of the FPS dealers were keen observers and could differentiate between FRKs and unfortified rice grains. However, none of them mentioned a difference in quality except an FPS dealer in Tamil Nadu.

"Cooked fortified rice becomes sticky on fermentation." (An FPS dealer, Tiruchirappalli district, Tamil Nadu)

"We do a manual check of fortified rice mostly while distributing them. Currently the quality of the fortified rice is good and there are no issues related to it." (An FPS dealer, Narmada district, Gujarat)

**Theme 3 –Experience in the distribution of fortified rice.** There was a continuous supply of fortified rice, and the demand for the rice was high.

"There is a continuous supply of fortified rice to beneficiaries across all talukas of Gadchiroli since the inception (Jan 2021). We never faced interruption of fortified rice supply to date." (An FPS dealer, Gadchiroli District, Maharashtra)

"There is a good demand and most often rice gets collected within a few days (2–3 days usually) from the 5th of every month." (An FPS dealer, Chandauli District, Uttar Pradesh)

**Theme 4 –Feedback from the beneficiaries.** When enquired about the feedback they received from the beneficiaries of the fortified rice, contrary to the DSO, the FPS dealers gave a mixed response.

"Few of us, as well as beneficiaries, are not happy with the quality of fortified rice. Many people avoid eating thicker rice being provided through FPS and therefore sell it in the

market at a lower price (at 10–12 Indian Rupees per Kg). To eat, either they grow (on their farms) or purchase thin rice from the market." (An FPS dealer, Gadchiroli District, Maharashtra)

"Currently there are no issues related to fortified rice. Earlier people used to say it as 'plastic rice' and used to complain about the changes in color (light pink) and the floating of kernels. But now, the feedback is all good." (An FPS dealer, Narmada district, Gujarat)

"The people consuming the fortified rice tell me that the taste of this rice is better than the rice we used to get before." (An FPS dealer, East Singhbhum District, Jharkhand)

## 4. Beneficiaries

The data collected from beneficiaries using the structured questionnaires revealed qualitative and quantitative (Table 1) components.

### Theme 1 –Quality of the fortified rice

Although the general perception of the quality of the fortified rice was good, the beneficiaries had some concerns as well.

"There is increased stickiness of cooked fortified rice." (A beneficiary from Gadchiroli district, Maharashtra)

"Rice distributed through FPS is thicker compared to rice purchased in the market." (A beneficiary from Gadchiroli district, Maharashtra)

"Some fortified rice kernels float and we throw them away while washing and cooking rice." (A beneficiary from Narmada district, Gujarat)

**Theme 2 –Experience about the IEC activities related to fortified rice.**   We could understand that IEC activities related to fortified rice were carried out in all the six districts in the six states. However, their perception of the IEC activities varied from state to state.

"I didn't know the rice was fortified until I saw the IEC activities." (A beneficiary from Narmada district, Gujarat)

"Govt officials, local leaders, and FPS shop owners should keep educating us on this." (A beneficiary of the Fortified Rice Program, Vijayanagaram District, Andhra Pradesh)

"Initially, we used to pick the chalky-white kernels from the rice and throw them away. But later we got information from the FPS dealer and from the posters that are displayed at the FPS shop about this rice and its benefits, and we started consuming it." (A beneficiary from East Singhbhum District, Jharkhand)

"The awareness activities were suboptimal. There are no banners (health education materials) at the DSO office, FPS shops, or in communities." (A beneficiary from the Gadchiroli district of Maharashtra)

**Theme 3 –Perceived changes in FR compared to unfortified rice.**   "FRKs have a different texture compared to the regular rice." (A beneficiary from the Gadchiroli district of Maharashtra)

"There is floating of rice kernels when it is being washed prior to the cooking, of about 10–20 grains at each cooking session." (A beneficiary from Vijayanagaram District, Andhra Pradesh)

"One or two white colored rice grains float over water while washing. The excess water is drained, and that water is fed to cattle or goats or thrown away." (A beneficiary from East Singhbhum District, Jharkhand)

**Theme 4 – Perceived adverse effects after the introduction of fortified rice.** Nobody complained about any kind of discomfort in terms of abdominal pain or diarrhea after they started eating fortified rice.

"We haven't had any adverse effects after beginning to consume fortified rice." (A beneficiary from Narmada district, Gujarat)

## 5. Results from direct observation of the beneficiaries

We observed decanting practice of water after cooking rice in three states. It is seen in households that do not use rice-cookers. Sufficient IEC activities are not undertaken to tackle this practice that can lead to loss of nutrients.

## 6. Results from structured questionnaire

All of the beneficiaries were consuming the fortified rice received through PDS either partially or completely. More than two-thirds (67.8%) of the total beneficiaries responded that at least one or two FRKs float while washing. However, all the respondents from Uttar Pradesh answered that the FRKs float while washing, while none of the respondents from Jharkhand experienced it. Among the beneficiaries who answered that FRKs float while washing, the proportion of people who throw away the floating kernels ranged from 0% in Uttar Pradesh to 90.5% in Maharashtra (mean = 27%). Out of 40 FRKs used to fortify 250 grams of rice, beneficiaries reported about floating of 4 or 5 FRKs (~10%).

More than 90% of respondents from all states did not feel any change in color of the fortified rice, while, except in Andhra Pradesh, most of the respondents in the other states did not feel a change in the texture of the fortified rice. While only 3.3% of the beneficiaries perceived a change in the way fortified rice cooks, 18.8% of them felt a change in the taste of fortified rice compared to the unfortified rice (Table 2). About 90% households in Vijayanagaram (Andhra Pradesh) were following the decanting practices (throwing away the excess water after cooking rice).

## Discussion

All the selected districts from six states in India (Maharashtra, Gujarat, Andhra Pradesh, Tamil Nadu, Uttar Pradesh, and Jharkhand) were supplying fortified rice to beneficiaries through PM POSHAN, ICDS, as well as PDS. The stakeholders such as the DSO, FCI/SFC managers, and the FPS suppliers were aware of their roles and responsibilities to distribute the fortified rice as envisaged in the government program. In three states, there is a practice of decanting the excess starchy water during cooking of fortified rice. Loss of iron, vitamin B12 & folic acid are reported if decanting of excess water is done. There was a good acceptability and compliance to the fortified rice consumption from the beneficiaries and there were no reported

**Table 2. Beneficiary responses regarding the access, availability, and utilization of fortified rice.**

| Variable/Question | Response | Total (N = 181) | Maharashtra (n = 30) | Gujarat (n = 28) | Andhra Pradesh (n = 28) | Tamil Nadu (n = 34) | Uttar Pradesh (n = 29) | Jharkhand (n = 32) |
|---|---|---|---|---|---|---|---|---|
| Type of ration card, n (%) | Regular PDS card | 125(69.1) | 19(63.3) | 11(39.3) | 26(92.9) | 31(91.2) | 22(75.9) | 16(50) |
| | AAY card | 48(26.5) | 11(36.7) | 9(32.1) | 2(7.1) | 3(8.8) | 7(24.1) | 16(50) |
| | others | 8(4.4) | 0(0.0) | 8(28.6) | 0(0.0) | 0(0.0) | 0(0.0) | 0(0.0) |
| Source of rice in previous one year, n (%) | From their farms | 0(0.0) | 0(0.0) | 8(28.6) | 0(0.0) | 0(0.0) | 0(0.0) | 0(0.0) |
| | From market | 0(0.0) | 0(0.0) | 8(28.6) | 0(0.0) | 0(0.0) | 0(0.0) | 0(0.0) |
| | PDS only | 86(47.51) | 17(56.7) | 14(50) | 28(100) | 18(52.9) | 0(0.0) | 9(28.1) |
| | Both PDS and Farm | 9(4.97) | 7(23.3) | 2(7.1) | 0(0.0) | 0(0.0) | 0(0.0) | 0(0.0) |
| | PDS and market | 85(46.96) | 5(16.7%) | 12(42.9) | 0(0.0) | 16(47.1) | 29(100%) | 23(71.9) |
| | PDS, farm, market | 1(0.55) | 1(3.3) | 0(0.0) | 0(0.0) | 0(0.0) | 0(0.0) | 0(0.0) |
| What do you receive in PDS?, n (%) | Rice | 28(15.5) | 0(0.0) | 0(0.0) | 28(100) | 0(0.0) | 0(0.0) | 0(0.0) |
| | Rice & wheat | 153(84.5) | 30(100) | 28(100) | 0(0.0) | 34(100) | 29(100) | 32(100) |
| Frequency of rice received through PDS in last one year, n (%) | Every month | 181(100) | 30(100) | 28(100) | 28(100) | 34(100) | 29(100) | 32(100) |
| How did you receive rice through PDS?, n (%) | Going to FPS | 153(84.5) | 30(100) | 28(100) | 0(0.0) | 34(100) | 29(100) | 32(100) |
| | Home delivery | 28(15.5) | 0(0.0) | 0(0.0) | 28(100) | 0(0.0) | 0(0.0) | 0(0.0) |
| How did you receive the rice supplied through PDS, n (%) | In a sack | 153(84.5) | 30(100) | 0(0.0) | 28(100) | 34(100) | 29(100) | 32(100) |
| | In own bag | 28(15.5) | 0(0.0) | 28(100) | 0(0.0) | 0(0.0) | 0(0.0) | 0(0.0) |
| Usage of PDS rice, n (%) | Consumed fully | 179(98.9) | 28(93.3) | 28(100) | 28(100) | 34(100) | 29(100) | 32(100) |
| | Consumed partially | 2(1.1) | 2(6.7) | 0(0.0) | 0(0.0) | 0(0.0) | 0(0.0) | 0(0.0) |
| Does FRKs float while washing? *, n (%) | Yes | 122(67.8) | 21(70) | 12(46.4)[#] | 27(96.4) | 33(97.1) | 29(100) | 0(0.0) |
| | No | 58(32.2) | 9(30) | 15(53.6) | 1(3.6) | 1(2.9) | 0(0.0) | 32(100) |
| What do you do with floating kernels?, n (%) | Throw away | 33(27) | 19(90.5) | 7(58.3) | 2(7.4) | 5(15.2) | 0(0.0) | 0(0.0) |
| | Use them for cooking | 89(73) | 2(9.5) | 5(41.7) | 25(92.6) | 28(84.8) | 29(100) | 0(0.0) |
| Color Change of FFR, n (%) | Yes | 4(2.2) | 0(0.0) | 1(3.6) | 0(0.0) | 0(0.0) | 1(3.4) | 2(6.25) |
| | No | 177(97.8) | 30(100) | 27(96.4) | 28(100) | 34(100) | 28(96.6) | 30(93.75) |
| Texture Change of FFR, n (%) | Yes | 41(22.7) | 5(16.7) | 2(7.1) | 27(96.4) | 6(17.6) | 0(0.0) | 1(3.1) |
| | No | 140(77.3) | 25(83.3) | 26(92.9) | 1(3.6) | 28(82.4) | 29(100) | 31(96.9) |
| Change in the way FFR cooks, n (%) | Yes | 6(3.3) | 0(0.0) | 0(0.0) | 0(0.0) | 0(0.0) | 0(0.0) | 6(18.75) |
| | No | 175(96.7) | 30(100) | 28(100) | 28(100) | 34(100) | 29(100) | 26(81.25) |
| Change in Taste of cooked FFR, n (%) | Yes | 34(18.8) | 1(3.3) | 0(0.0) | 26(92.9) | 7(20.6) | 0(0.0) | 0(0.0) |
| | No | 147(81.2) | 29(96.7) | 28(100) | 2(7.1) | 27(79.4) | 29(100) | 32(100) |
| Amount of rice received in kilograms through PDS last month, mean (SD) | | 22.63(6.15) | 17.97 (6.1) | 17.97 (6.1) | 16.07 (5.61) | 18.75 (6.18) | 20.85(6.18) | 16.38 (4.02) |
| Amount of PDS rice in kgs consumed last month, mean (SD) | | 22.63(6.15) | 16.97 (6.09) | 16.97 (6.09) | 16.07 (5.61) | 18.75 (6.18) | 20.85(6.18) | 16.38 (4.02) |

*One or two kernels float while washing; #In Gujarat, one person did not respond.

gastrointestinal adverse effects following the introduction of the iron and micronutrient forti-fied rice.

All the districts reported initial resistance from the public during the initial stage of the rice fortification program. Moreover, the media accentuated the problem by terming it (fortified rice) 'plastic rice' and 'China rice' as reported by the interviewees [28]. However, following various IEC activities carried out at the grassroot levels by the authorities, the perception of the public regarding fortified rice had completely changed and is now positive. The supply chain system of fortified rice through PDS was robust, although it varied from state to state. To our knowledge, this is the first study from India that has assessed the roll out of fortified rice through PDS as well as compliance of community individuals to its (fortified rice) utilization.

## Comparison of findings with other studies

Our findings are similar to the findings elsewhere in the world. The iron fortified rice in Philippines had acceptance ratings ranging from 6.9 to 7.4 ("like moderately" in the hedonic scale) [29]. A study done in Nepal found that most subjects could not distinguish between fortified and unfortified rice. They also found that the fortified rice scored 3.9 out of five in sensory qualities deeming it "acceptable" [30].

Our study shows that a few FRKs were floating while washing rice. The percentage of floating FRKs as responded by the beneficiaries ranged from 46.4% in Gujarat to 100% in Uttar Pradesh. Out of this, 27% of the respondents reported that they generally throw away the floating kernels instead of using them for cooking. The floating of kernels could be due to broken kernels or because of the higher temperatures used while drying the FRKs during the production (ideal temperature: below 65˚C). The practice of throwing FRKs had come down after health awareness activities from the health department. Additionally, government is taking all efforts to improve the texture, weight, color of FRK to make it more similar to regular rice grain. We also learned that the water used for washing the rice and the left-over water after cooking (decantation) was thrown away by most of the respondents. Such practices followed by families availing fortified rice can lead to loss of nutrients and therefore decrease the desired effects of the fortified rice [31]. Therefore, we suggest either using a rice cooker for cooking rice rather than cooking vessels or consuming the water used for cooking rice directly or indirectly. There must also be awareness campaigns to address this issue. These have been summarized in Table 3.

In this study, none of the beneficiary household have reported gastrointestinal side effects following intake of fortified rice. In contrast to our findings, previous studies have reported the gastrointestinal side effects such as abdominal pain and diarrhea (especially among

**Table 3. Possible threats to the success of fortification with suggested solutions.**

| Sl. No. | Possible threats to the success of distribution of fortified rice through PDS in India | Possible Solutions |
|---|---|---|
| 1. | Rejection of Fortified rice | • Improving the texture, weight, color, and density of FRK to make it similar to regular rice grain |
| 2. | Throwing away of FRK | • Improvement in density of the FRK<br>• IEC activities<br>• Regular monitoring of the FRK suppliers in adhering to the standard guidelines during the FRK manufacturing so that FRK appears like the regular rice grains. |
| 3. | Decantation of left-over water after cooking rice | • IEC activities to increase awareness not to decant the liquid.<br>• Using a rice cooker, if feasible. |

children) following the consumption of fortified rice as reported by Thankachan P et al [27]. The reason could be because of the differences in the form of iron used to fortify the rice. In the previous studies, which have reported the gastrointestinal adverse events following the intake of fortified rice, the rice was fortified with Ferrous sulphate compared to Ferric pyrophosphate (FPP, which is used in India). Ferrous sulphate is a soluble form of iron that generates free radicals and stimulate inflammatory cytokines. Whereas, micronized FPP (reduced to fine powders in the size of microns) is an insoluble form of iron and considered as safe to fortify the rice [32,33]. In addition to specific form of iron, consumption of rice fortified with both iron and zinc has reported to increase diarrhea/dysentery, possibly due to inhibition of iron absorption by zinc; the rice fortification program in India doesn't use zinc to fortify the rice.

## Current status of rice fortification in India

Currently, the GOI's strategy for food fortification with iron is 'targeted' fortification, which aims to increase the intake of specific subgroups of the population, rather than universal fortification [34]. In contrast, mass fortification, which adds one or more micronutrients to commonly consumed foods like cereals, milk, and condiments, is usually mandated, and regulated by the government sector [34]. The Food Safety & Standards Authority of India mandates the use of FPP (28–42.5 mg/kg) or sodium iron ethylene diamine tetra acetate Trihydrate (Na Fe EDTA 14–21.25 mg/kg) for fortification of rice with Iron in India. FPP is added at a higher level to account for its lower bioavailability [17]. If 35 mg of FPP is used for fortifying one kg rice, the estimated additional daily iron intakes through rice consumption are 0.9 mg/day among children aged 6–12 months, 5.9 mg/day among women of reproductive age, 6.0 mg/day among pregnant women, and 6.2 mg/day among adult men [35].

## Public health implication of the findings

Food fortification, if done as per standardized guidelines, is considered as one of the measures to reduce the burden of micronutrient deficiencies and improve health at community level. Food fortification can be considered as a temporary measure to control micronutrient deficiencies especially in vulnerable populations until more up-stream long term approaches such as diversification of diets are made available. The Copenhagen consensus of 2008 and 2012 found that micronutrient fortification and supplements to increase the nutrient intake were one of the most cost-effective investments with massive benefits [23,24]. Following this notion, in a majorly rice-consuming country like India, fortification of rice with iron and other micronutrients could be very beneficial if the acceptability, availability, and compliance are ensured. However, public health efforts to fortify rice (at national level in India) should be coupled with regular monitoring of dietary intakes, impact evaluation, adverse effects in different population segments, risk of overconsumption, development of biomarkers of excess intake, and long-term health effects. IEC posters must highlight the importance of not decanting the excess water after cooking rice. To inform policy decisions to scale up the distribution of fortified rice to national level, safety studies should be taken up for monitoring long term effects of fortified rice on the gut microbiome and inflammation. Prospective cohort studies are required, with detailed information about baseline and endline iron status with sufficient control for intake of other dietary components.

## Strengths and weaknesses of the study

The strengths of the study are: First, the study was conducted in six districts of different states of India covering North, West, South, and Central regions of India. Second, we employed a

mixed-method strategy, which included various data collection methods such as structured interviews, in-depth interviews, direct observation of various rice cleaning, cooking and consumption practices, along with quantitative data collection. This approach comprehensively helped us to understand the supply, distribution, acceptance, and various other issues related to the rice fortification program at beneficiary level. Third, we included diverse group of stakeholders involved in the distribution as well as utilization of fortified rice to get a complete view of the program from all perspectives. Fourth, all the interviews conducted were in-person interviews by a trained interviewers using a validated questionnaire and interview guide (informed by nutrition experts).

One of the weaknesses of the study was the response bias to interview questionnaires. It is possible that beneficiaries might have responded positively to intake of fortified rice due to fear of losing membership of PDS (if said no to fortified rice consumption). However, the possibility of this bias is low as all investigators conducted interviews in private place and ensured complete privacy of the participants to seek their response. Second, although the study was conducted in states from the different regions of India, caution should be exercised while generalizing the findings of the study to other parts of India (as the rice cooking and consumption practices varies across different states as well as regions).

## Conclusion

Across visited districts, there was a continuous supply of fortified rice in all talukas/blocks through a well-established system including health awareness activities. There was a good acceptability and compliance to distribution as well as intake of the iron fortified rice with no reported gastrointestinal adverse outcomes (following its intake). Findings of this study suggest a feasibility to conduct the effectiveness study to understand the effect of fortified rice on hemoglobin and iron storage at the community level to inform policy decision to scale up the fortified rice distribution at national level.

## Supporting information

**S1 Table. List of pilot districts where the centrally sponsored pilot scheme on fortification of rice and its distribution under public distribution system was implemented from 2020 to 2021.**
(DOCX)

**S1 File. Study questionnaires.**
(DOCX)

## Acknowledgments

We thank the district level officers, FCI/state Warehouse staff, fair price shop dealers and beneficiaries for their participation in the research. We thank the department of food and public distribution, Ministry of civil supplies, Government of India for the necessary approvals and permissions for conducting this research. We thank the department of public distribution in the states of Andhra Pradesh, Gujarat, Jharkhand, Maharashtra, Tamil Nadu, and Uttar Pradesh for giving necessary permissions to conduct this study.

## Author Contributions

**Conceptualization:** E. R. Nandeep, Hemant Mahajan, Challa Sairam, Hemalatha R., Samarasimha Reddy N.

**Data curation:** E. R. Nandeep.

**Formal analysis:** Hemant Mahajan, Mahesh Kumar Mummadi, Challa Sairam, Venkatesh K., Jayachandra Kadiyam, Sreenu Pagidoju, Venkata Raji Reddy, Hrusikesh Panda, Raghu Pullakandham, Subbarao M. Gavaravarapu.

**Funding acquisition:** Hemalatha R., Samarasimha Reddy N.

**Investigation:** Hemant Mahajan, Mahesh Kumar Mummadi, Challa Sairam, Venkatesh K., Jayachandra Kadiyam, Indrapal Meshram, Sreenu Pagidoju, Venkata Raji Reddy, Hrusikesh Panda, Raghu Pullakandham, Samarasimha Reddy N.

**Methodology:** Hemant Mahajan, Challa Sairam, Subbarao M. Gavaravarapu, Samarasimha Reddy N.

**Project administration:** E. R. Nandeep, Mahesh Kumar Mummadi, Challa Sairam, J. J. Babu Geddam, Samarasimha Reddy N.

**Validation:** Subbarao M. Gavaravarapu.

**Visualization:** Hemant Mahajan, Samarasimha Reddy N.

**Writing – original draft:** E. R. Nandeep, Hemant Mahajan, Challa Sairam, Venkatesh K., Jayachandra Kadiyam.

**Writing – review & editing:** Indrapal Meshram, Raghu Pullakandham, J. J. Babu Geddam, Subbarao M. Gavaravarapu, Hemalatha R., Samarasimha Reddy N.

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
