## [Decision Letter · Decision Letter 0]

8 Apr 2024

PGPH-D-23-02620

Implementation, delivery, and utilization of iron fortified rice supplied through public distribution system across different states in India: An exploratory mixed-method study

Dear Dr. Reddy N,

Thank you for submitting your manuscript to PLOS Global Public Health. After careful consideration, we feel that it has merit but does not fully meet PLOS Global Public Health’s publication criteria as it currently stands. Therefore, we invite you to submit a revised version of the manuscript that addresses the points raised during the review process.

The manuscript has been evaluated by two reviewers, and their comments are available below. Please could you revise your manuscript accordingly and provide detailed responses to the issues raised.

We look forward to receiving your revised manuscript.

Kind regards,

Johanna Pruller, Ph.D.

PLOS Staff Editor

Journal Requirements:

If you did not receive any funding for this study, please simply state: “The authors received no specific funding for this work.

3. Please send a completed 'Competing Interests' statement, including any COIs declared by your co-authors. If you have no competing interests to declare, please state "The authors have declared that no competing interests exist". Otherwise please declare all competing interests beginning with the statement "I have read the journal's policy and the authors of this manuscript have the following competing interests:"

4. Please provide separate figure files in .tif or .eps format only and remove any figures embedded in your manuscript file. Please also ensure all files are under our size limit of 10MB.

5. In the online submission form, you indicated that "The data is available on request to corresponding author". All PLOS journals now require all data underlying the findings described in their manuscript to be freely available to other researchers, either 1. In a public repository, 2. Within the manuscript itself, or 3. Uploaded as supplementary information.

6. We have noticed that you have uploaded Supporting Information files, but you have not included a list of legends. Please add a full list of legends for your Supporting Information files after the references list.

Figures 1 & 2: Please confirm (a) that you are the photographer; or (b) provide written permission from the photographer to publish the photo(s) under our CC-BY 4.0 license.

Additional Editor Comments (if provided):

Reviewers' comments:

Reviewer's Responses to Questions

**Comments to the Author**

1. Does this manuscript meet PLOS Global Public Health’s publication criteria? Is the manuscript technically sound, and do the data support the conclusions? The manuscript must describe methodologically and ethically rigorous research with conclusions that are appropriately drawn based on the data presented.

Reviewer #1: Yes

Reviewer #2: Yes

2. Has the statistical analysis been performed appropriately and rigorously?

Reviewer #1: N/A

Reviewer #2: N/A

3. Have the authors made all data underlying the findings in their manuscript fully available (please refer to the Data Availability Statement at the start of the manuscript PDF file)?

Reviewer #1: Yes

Reviewer #2: No

4. Is the manuscript presented in an intelligible fashion and written in standard English?

Reviewer #1: Yes

Reviewer #2: Yes

5. Review Comments to the Author

Reviewer #1: The strategy evaluated is undoubtedly important. However, it has been published that social group is important for the prevalence of anemia in women [Sharif, N., Das, B., & Alam, A. (2023). Prevalence of anemia among reproductive women in different social group in India: Cross-sectional study using nationally representative data. PloS one, 18(2), e0281015. ], undernutrition is related to anemia in children [Stiller, C. K., Golembiewski, S. K. E., Golembiewski, M., Mondal, S., Biesalski, H. K., & Scherbaum, V. (2020). Prevalence of Undernutrition and Anemia among Santal Adivasi Children, Birbhum District, West Bengal, India. International journal of environmental research and public health, 17(1), 342. https://doi.org/10.3390/ijerph17010342], and anemia is disproportionately concentrated in low socioeconomic groups in low-income and middle-income countries [Balarajan, Y., Ramakrishnan, U., Ozaltin, E., Shankar, A. H., & Subramanian, S. V. (2011). Anaemia in low-income and middle-income countries. Lancet (London, England), 378(9809), 2123–2135. https://doi.org/10.1016/S0140-6736(10)62304-5]. Maybe it is not polite to remember the reality: poverty leads to anemia. Iron deficiency anemia in India is an old problem [Anand, T., Rahi, M., Sharma, P., & Ingle, G. K. (2014). Issues in prevention of iron deficiency anemia in India. Nutrition (Burbank, Los Angeles County, Calif.), 30(7-8), 764–770. https://doi.org/10.1016/j.nut.2013.11.022]. Onyeneho considers that "Combining nutritional supplementation and food-fortification programmes with reduction in maternal anemia and family poverty may yield optimal improvement of childhood anemia in India." [Onyeneho, N. G., Ozumba, B. C., & Subramanian, S. V. (2019). Determinants of Childhood Anemia in India. Scientific reports, 9(1), 16540. https://doi.org/10.1038/s41598-019-52793-3]. Authors could explain a little bit more why they did not profounder regarding poverty. Poverty is related with low education, undernutrition, an some other problems previous to develop anemia. It is a strength the study was conducted in six districts of different states, but it would be better some kind of stratification of population. It is known that one of the weaknesses of qualitative studies is the response bias to interview questionnaires, so authors could use population hemoglobin registries to correlate with potential benefits of the utilization of iron fortified rice supplied by government.

Reviewer #2: This paper reports on interviews and surveys from a range of stakeholders in India’s public distribution of fortified rice. The program is ambitious and the stakeholders seem largely positive about its success. This seems like valuable information to get into the published literature, and I offer some suggestions for how the authors might make minor revisions to make their work more accessible to a wider range of readers.

Line 74: Folic acid --- also very valuable, especially for prevention neural tube defects, and perhaps best delivered through fortified foods. Worth emphasizing more.

Line 90: did the Cochrane review report a confidence interval for the mean difference of 109? If so, you should include it, too.

Line 125: you could explain here what exactly a “fair price shop” is, for readers who are not familiar (you provide a bit more detail on Line 341-342, but I would be interested to know even more, and sooner)

Line 129: conducting all of these interviews in five days seems impossible! Are you sure the May 1 to 5, 2023 time period is correct?

Line 142 / Fig 1: I found this figure distracting. Perhaps a table would be a more effective way to share this information.

Line 143-147: consider specifying here which were qualitative interviews and which were quantitative. If I understand correctly, the 181 beneficiaries provided quantitative responses to a structured interview while all the rest were unstructured interviews with qualitative analyses.

Line 150-157: I would be interested in a summary table and/or some examples of these constructs.

Line 171-172: I was surprised that you needed to take special care to avoid interviewing the same respondents for the qual and quant parts. Did I misunderstand something?

Line 182-183: I suggest you include your questionnaires as supplementary material.

Line 185: here is says you had 36 unstructured interviews, but in Figure 1 you say 52. I think these should be the same number.

Figure 2: I don’t know if this figure add much, I found it distracting.

Line 256: the IEC activities seem important! Do you have more detail from your qual analysis that you can include about them?

Line 441: I would like more information about your approach to direct observation. What was the protocol? How many subjects were observed? How did you have time to do this in only five days of data collection?

Line 469: Table 1 shows a high level of “clustering” by location. I would be interested to know more about the reasons behind this, and it could be a more useful way to organize the information you currently have presented in Table 1.

Line 493: what is the scale of acceptance rating? Is 6.9 high acceptance?

Line 500: it seems like your work has identified at least three important threats to success in fortification: (1) rejecting the product (2) picking out the FRK (3) decanting the liquid. You should include a table or figure to emphasize these. Are there other important threats that you discovered?

Line 519: I am not familiar with the term of art “micronized FPP”. Consider including a reference for other readers who are unfamiliar and want to learn more.

6. PLOS authors have the option to publish the peer review history of their article (what does this mean?). If published, this will include your full peer review and any attached files.

**Do you want your identity to be public for this peer review?** For information about this choice, including consent withdrawal, please see our Privacy Policy.

Reviewer #1: **Yes: **Jorge Alberto Álvarez Díaz

Reviewer #2: No

While revising your submission, please upload your figure files to the Preflight Analysis and Conversion Engine (PACE) digital diagnostic tool, https://pacev2.apexcovantage.com/. PACE helps ensure that figures meet PLOS requirements. To use PACE, you must first register as a user. Registration is free. Then, login and navigate to the

---

## [Decision Letter · Decision Letter 1]

5 Jul 2024

Implementation, delivery, and utilization of iron fortified rice supplied through public distribution system across different states in India: An exploratory mixed-method study

PGPH-D-23-02620R1

Dear Dr Reddy N,

We are pleased to inform you that your manuscript 'Implementation, delivery, and utilization of iron fortified rice supplied through public distribution system across different states in India: An exploratory mixed-method study' has been provisionally accepted for publication in PLOS Global Public Health.

Best regards,

Julia Robinson

Executive Editor

Reviewer Comments (if any, and for reference):

Reviewer's Responses to Questions

**Comments to the Author**

1. If the authors have adequately addressed your comments raised in a previous round of review and you feel that this manuscript is now acceptable for publication, you may indicate that here to bypass the “Comments to the Author” section, enter your conflict of interest statement in the “Confidential to Editor” section, and submit your "Accept" recommendation.

Reviewer #1: All comments have been addressed

Reviewer #2: All comments have been addressed

2. Does this manuscript meet PLOS Global Public Health’s publication criteria? Is the manuscript technically sound, and do the data support the conclusions? The manuscript must describe methodologically and ethically rigorous research with conclusions that are appropriately drawn based on the data presented.

Reviewer #1: Yes

Reviewer #2: Yes

3. Has the statistical analysis been performed appropriately and rigorously?

Reviewer #1: Yes

Reviewer #2: Yes

4. Have the authors made all data underlying the findings in their manuscript fully available (please refer to the Data Availability Statement at the start of the manuscript PDF file)?

Reviewer #1: Yes

Reviewer #2: Yes

5. Is the manuscript presented in an intelligible fashion and written in standard English?

Reviewer #1: Yes

Reviewer #2: Yes

6. Review Comments to the Author

Reviewer #1: I consider the requirements were fully fulfill.

Reviewer #2: This revision has addressed all of my comments and concerns.

7. PLOS authors have the option to publish the peer review history of their article (what does this mean?). If published, this will include your full peer review and any attached files.

**Do you want your identity to be public for this peer review?** For information about this choice, including consent withdrawal, please see our Privacy Policy.

Reviewer #1: **Yes: **Jorge Alberto Álvarez Díaz

Reviewer #2: No
